



# Transport pathways of carbon monoxide from Indonesian fire pollution to a subtropical high-altitude mountain site in western North Pacific

Saginela Ravindra Babu[1*], Chang-Feng Ou-Yang[1], Stephen M. Griffith[1], Shantanu Kumar Pani[1],

Steven Soon-Kai Kong[1], and Neng-Huei Lin[1,2*]

[1]Department of Atmospheric Sciences, National Central University, Taoyuan 32001, Taiwan.

[2]Center for Environmental Monitoring and Technology, National Central University, Taoyuan

32001, Taiwan.

Correspondence to: S. Ravindra Babu (baburavindra595@gmail.com) and Neng-Huei Lin
(nhlin@cc.ncu.edu.tw).

**Abstract:** Dry conditions associated with El Niño and a positive Indian Ocean Dipole (IOD) are known to have caused major fire pollution events and intense carbon emissions over a vast spatial expanse of Indonesia in October 2006 and 2015. During these two events, a substantial increase in carbon monoxide (CO) mixing ratio was detected by in-situ measurements at Lulin Atmospheric Background Station (LABS, 23.47°N 120.87°E, 2,862 m ASL) in Taiwan, the only background station in the subtropical western North Pacific region. Compared to the long-term October mean (2006-2021), CO was elevated by ~47.2 ppb (37.2%) and ~36.7 ppb (28.9%) in October 2006 and 2015, respectively. This study delineates plausible pathways for CO transport from Indonesia to LABS using MOPITT CO observations and MERRA-2 reanalysis products (winds and geopotential height (GpH)). Two simultaneously occurring transport pathways were identified: (i) horizontal transport in the free troposphere and (ii) vertical transport through the Hadley circulation (HC). The GpH analysis of both events revealed the presence of a high-pressure anticyclone over the northern part of the South China Sea (SCS), which played an important role in the free tropospheric horizontal transport of CO. In this scenario, CO in the free troposphere is transported on the western edge of the high-pressure system and then driven by subtropical westerlies to LABS. Simultaneously, uplifted CO over Indonesia can enter the HC and transfer to subtropical locations such as LABS. The vertical cross-section of MOPITT CO and MERRA-2 vertical pressure velocity supported the transport of CO through the HC. Further, the results revealed a distinct HC strength in two events (higher in 2006 compared to 2015) due to the





different El Niño conditions. Overall, the present findings can provide some insights into understanding the regional transport of pollution over Southeast Asia and the role of climate conditions on transport pathways.

**Keywords:** Indonesian fire pollution; Carbon monoxide; Lulin Atmospheric Background Station; Hadley circulation

## 1. Introduction

Fire activity over Southeast Asia (SEA), particularly over the Maritime Continent (MC, including Indonesia), is a severe environmental problem that causes widespread regional pollution in the lower troposphere and impacts atmospheric chemistry, air quality, and climate at regional to global scales. Over the MC, fires occur predominately in the dry season (August to October) and particularly during the periods of drought, often associated with the positive phase of El Niño-Southern Oscillation (ENSO) events (Duncan et al., 2003a; van der Werf et al.,2008, 2017; Field et al., 2009, 2016). A recent study has also highlighted the role of the Indian Ocean Dipole on MC fire activity (Pan et al., 2018). For example, dry conditions associated with the extreme 2015/16 El Niño and weak 2006/07 El Niño events led to increased fire activity over Indonesia and the wider MC (van der Werf et al.,2008; Chandra et al., 2009; Nassar et al., 2009; Huijnen et al., 2016; Field et al., 2016). Due to these intense fires, an enormous amount of carbon emissions was released into the atmosphere in the form of carbon dioxide ($CO_2$), carbon monoxide (CO), and methane ($CH_4$) (Huijnen et al., 2016; Field et al., 2016; Parker et al., 2016; Heymann et al., 2017). These two Indonesian fire events and the associated impacts on carbon emissions, trace gas and aerosol composition, and air quality has been extensively discussed in the literature (Chandra et al., 2006; Logan et al., 2008; Chandra et al., 2009; Nassar et al., 2009; Huijnen et al., 2016; Field et al., 2016; Heymann et al., 2017; Ravindra Babu et al., 2019). For example, the fire carbon emissions during September-October 2015 over Maritime SEA were the largest since 1997 (Huijnen et al., 2016). By using Greenhouse gases Observing SATellite (GOSAT) data, Parker et al. (2016) reported the strong enhancement of $CO_2$ and $CH_4$ over the Indonesian region.

CO is a significant emission from the combustion of fossil fuels and biomass (forest and savanna fires, biofuel use, and waste burning) and is widely used as a tropospheric tracer for these sources (Ou-Yang et al., 2014; Pani et al., 2019). Inter-annual variability of CO in the tropics and



sub-tropics is largely linked to year-by-year changes in biomass burning (BB) emissions.
Indonesian fires often emit large quantities of CO by incomplete combustion associated with the
occurrence of peat fire pollution. Although CO is not a direct greenhouse gas (GHG), it does have
a global warming potential due to its chemical reactions in the atmosphere. For instance, CO
oxidation produces $CO_2$ and indirectly ozone ($O_3$), both of which are GHGs, and depletes hydroxyl
radical (OH) concentrations, thus extending the lifetime of $CH_4$, another GHG in the atmosphere
(IPCC, 2013). The lifetime of CO in the free troposphere is ∼ two months, thus can be a tracer
from polluted upwind regions to remote downwind areas (Cooper et al., 2012). Some of the studies
reported the influence of Indonesian fire activity and the transport of CO from Indonesia to the
Indian Ocean, Southern Pacific, and western Pacific Ocean (Matsueda and Inoue, 1999; Pochanart
and Akimoto, 2003; Nara et al., 2011; Matsueda et al., 2002, 2019). However, the underlying
transport mechanisms sending this fire pollution to downwind northern hemisphere subtropical
locations, particularly transport to high-altitude background locations in the western north Pacific
are still unclear.

As shown in **Figure 1**, the Lulin Atmospheric Background Station (LABS, 23.47°N

120.87°E, 2862 m ASL) located in central Taiwan was constructed in 2006 and is the only high-
altitude background station in the western Pacific region for monitoring the long-term variability
of atmospheric compositions and also studying the influence of continental outflow and long-range
transported pollution (Lin et al., 2013; Ou-Yang et al., 2014, 2022; Ravindra Babu et al., 2022).
The LABS is often found within the free troposphere, making it an ideal site for measuring long-
range transport of air pollutants, complementing the global network of the Global Atmospheric
Watch (GAW) in the East Asia region where no other high-altitude background station is available
(Ou-Yang et al., 2014, 2022). In the framework of Seven South-East Asian Studies (7-SEAS, Reid,
et al., 2013; Lin et al., 2013; Wang et al., 2015), several studies at LABS have reported on the
long-range transport of northern peninsular Southeast Asia (PSEA) BB pollutants to Taiwan
through the low-level jet (LLJ) and the related impacts on air quality and chemistry over Taiwan
(Ou-Yang et al., 2012, 2014; Lin et al., 2013; Chuang et al., 2016; Chi et al., 2016; Tsay et al.,
2016; Hsiao et al., 2016; Lin et al., 2017; Park et al., 2019; Pani et al., 2016, 2019; Huang et al.,
2019; Huang et al., 2020; Ravindra Babu et al., 2022). However, to date, no studies have shown
the potential influence of Indonesian fire activities on LABS measurements and the BB pollution
from Indonesian fires reaching LABS. Surprisingly, the extensive fire events in 2006 and 2015



allowed us to track CO concentrations from the Indonesian peat fires to LABS in Taiwan. By
combining in-situ and satellite CO measurements and large-scale circulation parameters from
reanalysis products, we identified plausible transport pathways from Indonesia to LABS.

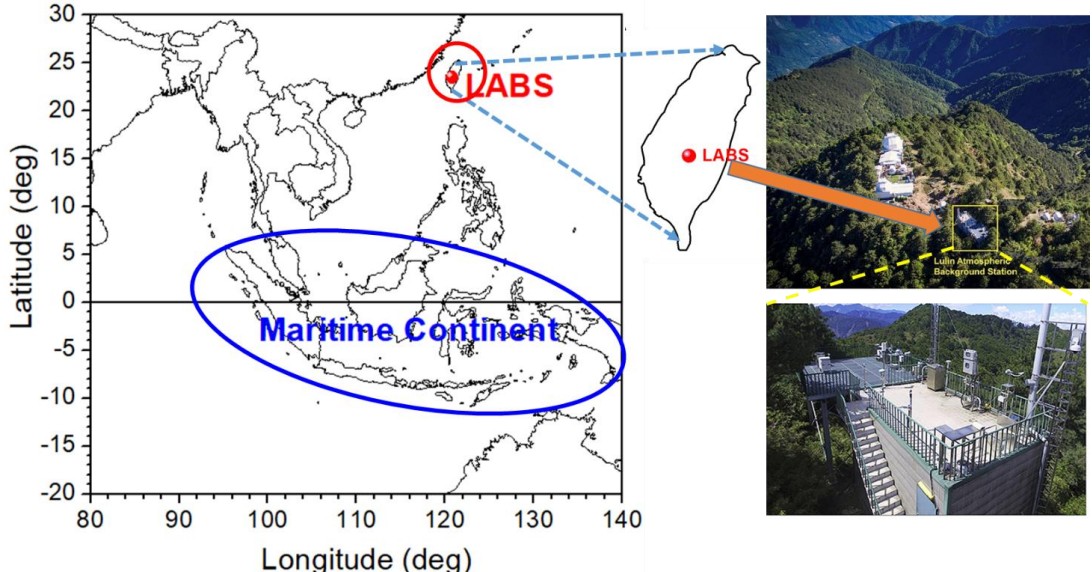


**Figure 1**. Geographic location of the Maritime Continent and Lulin Atmospheric Background
Station (LABS, 23.47°N 120.87°E, 2862 m ASL), Taiwan.
**2. Data and methodology**
**2.1 In-situ measurements**

CO mixing ratios were measured by a nondispersive infrared (NDIR) analyzer (APMA-

360, Horiba, Japan) at LABS. Hourly averages of the 6-s data were analyzed in this study. The
detection limit of the NDIR is ~20 ppb (1σ) (Zellweger et al., 2009); more details about CO
measured at LABS can be found in Ou-Yang et. (2014). The magnitude of the CO concentration
enhancement in 2006 and 2015 above the long-term background was determined by comparing a
16-year average (2006-2021) of October CO data at LABS. We obtained the percentage change in
CO relative to the respective background using Equation 1:

$$\text{Relative change in percentage} = \left(\frac{x_{i} - \bar{x}}{\bar{x}}\right) \times 100 \qquad (\text{Eq. 1})$$



where $x_i$ represents the monthly mean of October in 2006 and 2015, and $\bar{x}$ is the corresponding
monthly long-term mean calculated using the data from 2006 to 2021 (Ou-Yang et al., 2014).

**2.2 Satellite measurements**

CO observations from the Measurement of Pollution in the Troposphere (MOPITT, version
8) instrument were also utilized in this study (Worden et al., 2010; Deeter et al., 2019). MOPITT
is a multi-channel Thermal InfraRed (TIR) and Near InfraRed (NIR) instrument operating onboard
sun-synchronous polar-orbiting NASA Terra satellite. V8 CO products, consisting of a CO profile
at ten pressure levels, have been validated; more details about the retrieval algorithm, validation,
and the uncertainties of MOPITT CO can be found in Deeter et al. (2019). In addition to the
MOPITT measurements, we utilized CO from the Atmospheric Infrared Sounder (AIRS) on the
NASA Aqua satellite, which provides CO at different vertical levels twice daily and near-global
coverage. AIRS uses wavenumbers 2,183-2,200 cm$^{-1}$ (4.58-4.5 μm) for retrieving CO (McMillan
et al., 2005). Version 8, level 3 CO product, available at 1° ×1° resolution at various pressure
levels, was utilized in the present study. AIRS data were downloaded from the following website
https://disc.gsfc.nasa.gov/datasets/AIRS3STM_7.0 (AIRS project., 2019). AIRS sensitivity to CO
is broad and optimal in the mid-troposphere between approximately 300 and 600 hPa (Warner et
al., 2007; Warner et al., 2013; AIRS project., 2019). CO retrievals have a bias of 6-10% between
900 hPa and 300 hPa with a root mean square error of 8-12 % (McMillan et al., 2011).
Apart from MOPITT and AIRS CO data, we used Moderate Resolution Imaging
Spectroradiometer (MODIS) collection 6.1 daily active fire hot spot data from 2006–2020 over
Indonesia (Giglio et al., 2016).

**2.3 MERRA-2 Reanalysis products**

Modern-Era Retrospective Analysis for Research and Applications, version 2 (MERRA-2)
monthly mean geopotential height (GpH) wind vectors (zonal and meridional) and vertical
pressure velocity (omega) during the study period were utilized. MERRA-2 is the latest
atmospheric reanalysis data produced by the NASA Global Modeling and Assimilation Office
(GMAO) (Gelaro et al., 2017). The horizontal resolution of MERRA-2 reanalysis is 0.5º × 0.625º.
MERRA-2 data are available online through the NASA Goddard Earth Sciences Data Information
Services Center (GES DISC; https://disc.gsfc.nasa.gov/, last access: 11 September 2022).





## 3. Results and Discussion

### 3.1 Higher CO mixing ratios in October 2006 and 2015 at LABS

**Figure 2** summarizes the inter-annual variations of CO in October observed at LABS along with MODIS active fire counts over Indonesia and the observed Niño 3.4 and the IOD index values, which helped to motivate this study. The highest CO mixing ratios for this period were observed in 2006 and 2015, well over the long-term means of 132.1±23.3ppb when including all points and 126.8±19.6 ppb when excluding 2006 and 2015. A significant enhancement of CO, over the latter mean calculation, of more than 47.2 ppb (37.2%) in 2006 and 36.7 ppb (28.9%) in 2015 was observed, with the value in 2006 (2015) more significant than the ±2σ (±1σ) standard deviation of the long-term mean (**Table 1**). Higher CO mixing ratios in 2006 and 2015 at LABS were also evident from the MOPITT and AIRS satellite measurements obtained over a 1-degree radius around the LABS location (**Fig. 3**).

Unprecedented CO values in 2006 and 2015 at LABS could be due to the transport of CO from large-scale forest fires that were intense during the same period in the Indonesian region. It is clear from **Figure 2**, that the higher values of CO at LABS in 2006 and 2015 coincided with more intense fire activity over Indonesia along with warm phases of ENSO and IOD (**Fig. 2c and 2d**), which have been extensively studied due to the induced drought conditions in those years (Field et al., 2016; Huijnen et al., 2016; Pan et al., 2018). Several studies have reported on the impact of the intense BB in 2006 and 2015 on the release of significant carbon emissions and the air quality over the wider Equatorial Asian region (Logan et al., 2008; Chandra et al., 2009; Field et al., 2016; Huijnen et al., 2016; Ravindra Babu et al., 2019). The enhanced CO values from the 2006 and 2015 events at LABS in the present study complement the findings of Matsueda and Inoue (1999) in the case the of 1997 El Niño event and Nara et al. (2011) in the case of 2006 El Niño event. However, the impact on CO at LABS occurred significantly further north of the source region than in either of the aforementioned studies. Based on aircraft measurements, Matsueda and Inoue (1999) reported the enhancement of $CO_2$, CO, and $CH_4$ in the upper troposphere (at 9-12 km) over the South China Sea (SCS) during October 1997 Indonesian fire event. However, this large CO increase appeared only over the SCS west of Kalimantan and not in the subtropics between 10°N and 26°N. Nara et al. (2011) reported a substantial increase in CO mixing ratios over the Western Tropical Pacific Ocean (between 15°N and the Equator) by shipboard observations



routinely operated between Japan and Australia and New Zealand during October and November
of 2006. Similarly, Pochanart and Akimoto (2003) also reported the influence of the 1997
Indonesian fire event on CO enhancement at the rural station Srinakarin (14°220N, 99°070E, 296
m above sea level) in Thailand.

In addition, due to La Niña and the negative phase IOD, the fire activity in Indonesia during

2016 was much less intense than in 2006 and 2015 (**Fig. 2c** and **2d**). Interestingly, CO at LABS
during 2016 exhibited the lowest October values in the entire data period, ~39.8 ppb (31.4%) lower
than the long-term October mean (2006-2021). It is well known that the major sources of CO at
LABS are BB from peninsular SEA in spring and industrial emissions from continental Asia in
winter (Ou-Yang et al., 2014; Pani et al., 2019; Ravindra Babu et al., 2022; Ou-Yang et al., 2022).
However, October is a transition month from the summer to winter at LABS, when air masses can
still arrive from the Pacific Ocean. Our analysis (**Fig. 2**) suggests that the extensive fires that
occurred during the 2006 and 2015 El Niño events over Indonesia may have yielded the
unprecedented CO mixing ratios at LABS in October of those years.  Combined El Niño and IOD-
related changes in the large-scale dynamics and circulations may have promoted CO emissions
from Indonesian fires to transport to LABS.



**Figure 2.** Inter-annual variations in October of the (a) monthly median of CO, (b) percentage change in CO from the long-term mean at LABS, (c) MODIS (Moderate Resolution Imaging Spectroradiometer) total active fire counts (only fires tagged with >30 % confidence) over





Indonesia, (d) sea surface temperature index for Niño 3.4 (magenta) and Indian ocean dipole
(black) during 2006 to 2021.


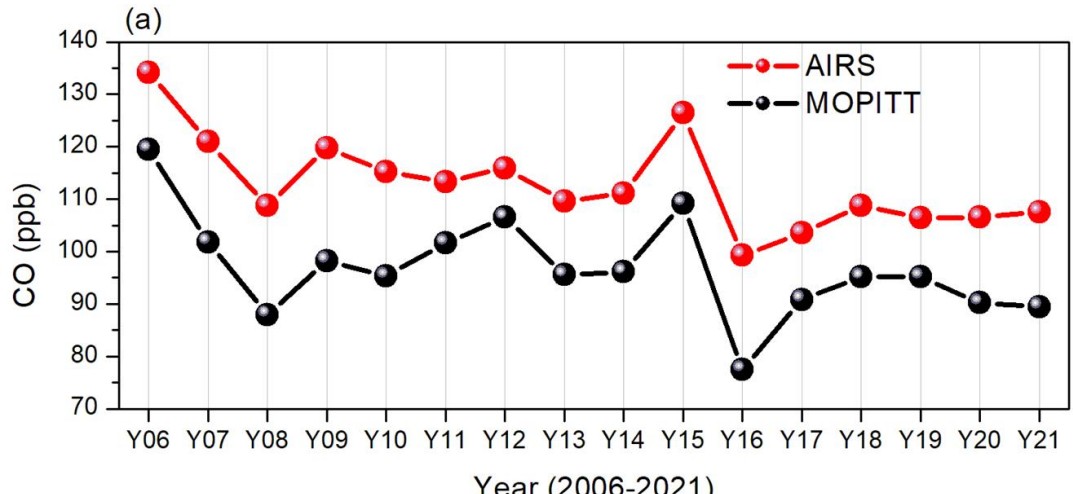

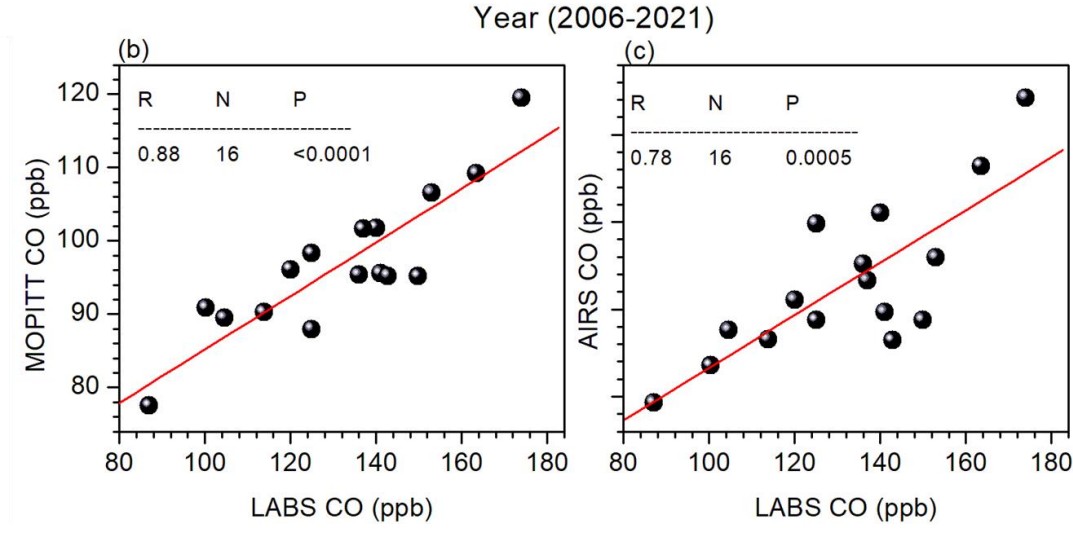


**Figure 3.** (a) MOPITT (black) and AIRS (red) satellite observed CO mixing ratios within the 1-
degree radius around the LABS location, (b) correlation plot between in-situ CO at LABS and
MOPITT CO, and (c) correlation plot between in-situ CO at LABS and AIRS CO in October
month during 2006 to 2021. (R is the correlation coefficient; N is the sample size; P is the
significance value)




To confirm the impact of Indonesian fire pollution on LABS CO, we further checked the
spatial distribution of CO in 2006 and 2015 from the MOPITT satellite CO observations. An inter-
comparison between October monthly mean CO at LABS (2006-2021) and MOPITT and AIRS
CO data within the 1-degree radius around the LABS location yielded correlation coefficients of
0.88 and 0.78 ($p<0.01$), respectively (**Fig. 3**). We then used the MOPITT satellite data to track the
spatial and vertical CO changes in October 2006 and 2015; first, we examined the distribution of
the CO anomalies at free tropospheric heights in those years. **Figure 4** shows these anomalies
compared to the long-term mean (2001-2021) at 700 hPa and 500 hPa, revealing extensive
enhancements of CO mixing ratios over most of equatorial Asia in 2006 and 2015. **Figure 4**
indicates that CO from the Indonesian fires affected both the Indian Ocean to the west and South
Pacific and the northern Pacific to the east. Furthermore, these outflows of CO split northwestward
into the Bay of Bengal and northeastward into the western North Pacific. It is also worth noting
that the anomalies were significantly higher at 500 hPa than 700 hPa. Elevated CO is visible in the
Taiwan region at 700 hPa and 500 hPa in both years. This further provides a clear signature of the
impact of Indonesian fire activity on enhanced CO in 2006 and 2015 at LABS. Overall, from
**Figure 4**, MOPITT CO data shows the Indonesia fires transported CO vertically and horizontally
in all directions. We further investigated the associated dynamics and large-scale circulations
supporting the transport of Indonesian pollution to LABS.



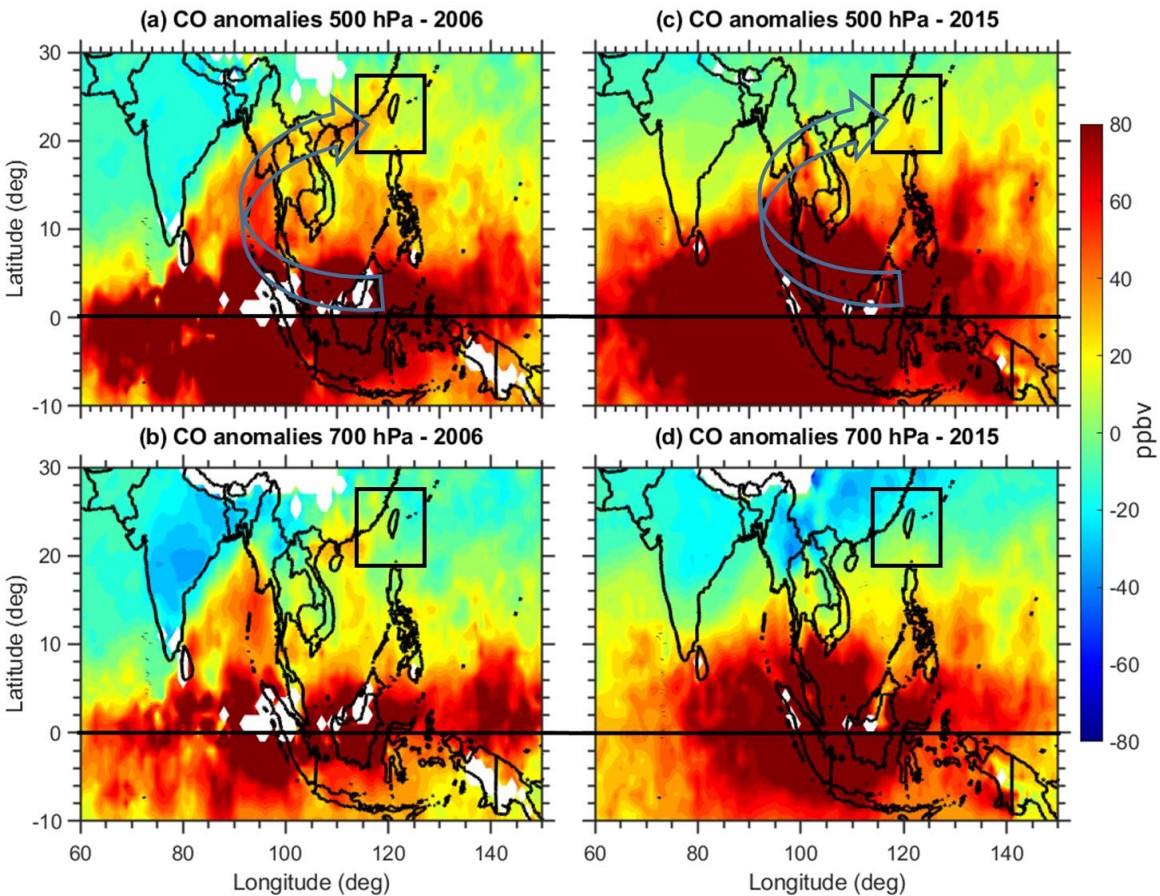

**Figure 4.** Monthly mean CO anomalies obtained from MOPITT satellite observations (a) at 500 hPa and (b) at 700 hPa during October 2006. Subplots (c) and (d) are the same as subplots (a) and (b) but for October 2015, respectively. The anomalies are obtained by subtracting the 2006 and 2015 data from the long-term mean of MOPITT CO data from 2001-to 2021.

### 3.2 Role of large-scale dynamics and atmospheric circulations

Large-scale dynamics and circulations can play a crucial role in transporting Indonesian pollution to long-distance downwind regions (Bowman, 2006; Nara et al., 2011; Matsueda et al., 2019). To understand the plausible mechanisms behind the transport of Indonesian fire pollution to LABS, we further examined the MERRA-2 reanalysis of geopotential height (GpH) and wind distribution in 2006 and 2015. The spatial distribution of GPH at two pressure levels (700 and 500 hPa) in both events is shown in **Figure 5**. The GpH and wind vectors in the two event years



exhibited quite different patterns in relation to a high-pressure system over the northern parts of
the SCS. A high-pressure anti-cyclonic circulation center extended from the Indo-China Peninsula
to the SCS in October 2006 with LABS located precisely on the eastern edge of the anticyclone.
In 2015, the anticyclone extended from the Indo-China Peninsula to the western North Pacific
region and over Taiwan.

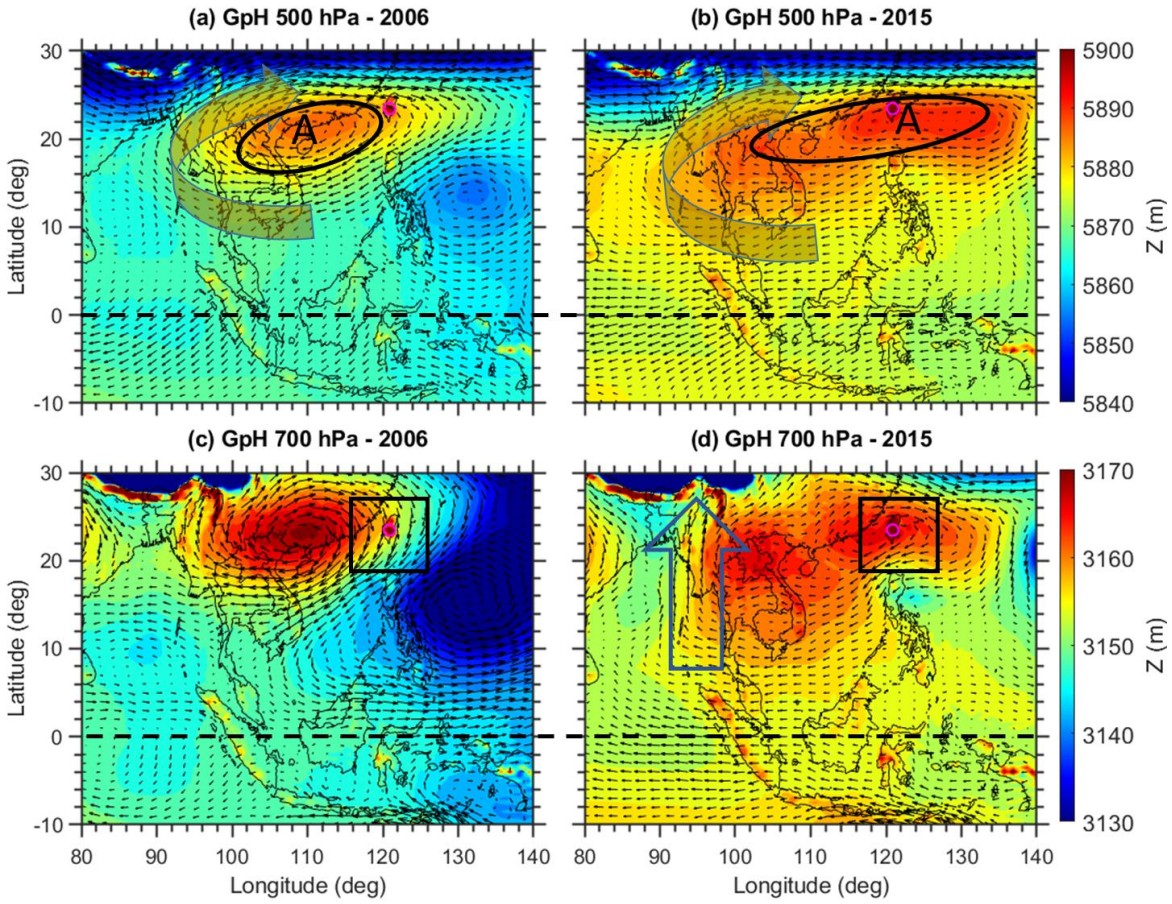


**Figure 5.** Monthly mean Geopotential height (GpH) obtained from MERRA-2 reanalysis (a) at
500 hPa and (b) at 700 hPa during October 2006. Subplots (c) and (d) are the same as subplots (a)
and (b) but for October 2015.

During both event years, strong southerlies at 500 hPa were evident due to the high-

pressure anticyclone system in the northern SCS. It is assumed that the northern edge of the
Indonesian fire pollution plume can be carried out by the southerlies and around the western edge





of the high-pressure anti-cyclone over SCS. An apparent merging of the southerlies from the
equator with the subtropical westerlies in the northern PSEA region subsequently led to the
transport of CO to downwind LABS. Overall, in both events, there was a significant anticyclone
over the SCS. El Niño and the positive IOD-induced high-pressure anticyclone over SCS
strengthen the southerlies from the equator, consequently bringing higher amounts of CO to LABS.
We further investigated the vertical pressure velocity (omega) behavior in both events (**Fig. 6**),
where negative (positive) values represent upward (downward) winds. Significant upward wind in
both events was evident over equatorial MC, while vertical pressure velocity over Taiwan and
surrounding regions at both pressure levels were mostly downwards in 2006 and 2015. The
presence of a downwind will provide downward transport of any pollutant presence in the upper
troposphere over that region. Also, the downward wind was relatively higher in 2006 compared to
2015. The center of the downward wind was shifted eastwards in the western North Pacific in
2015. The distinct behavior of vertical pressure velocity at the LABS region during the two events
might be due to the associated climate conditions in the two periods; more discussion will be
provided in section 3.4.

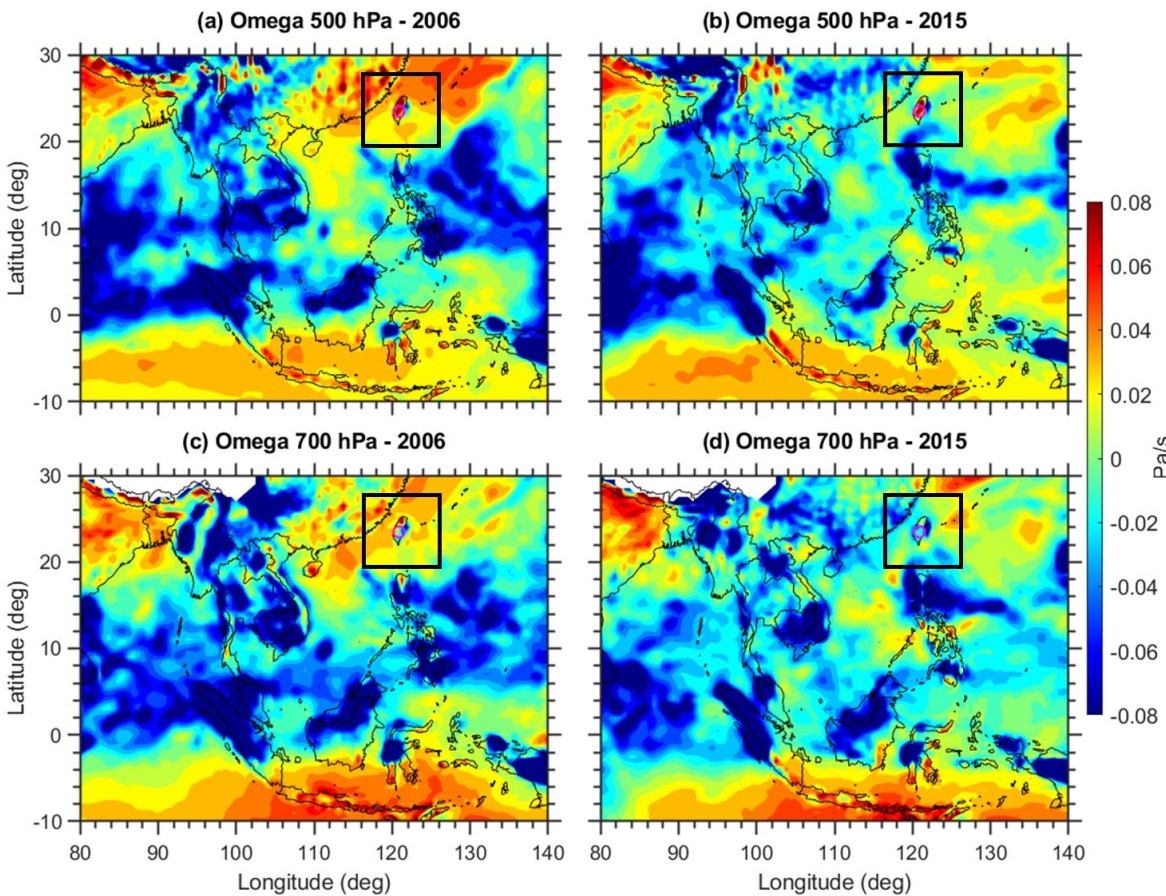


**Figure 6**. Monthly mean vertical pressure velocity obtained from MERRA-2 reanalysis (a) at 500 hPa and (b) at 700 hPa during October 2006. Subplots (c) and (d) are the same as subplots (a) and (b) but for October 2015.

We further showed CO deviations at both pressure levels in October 2016 when there was
very low fire activity in Indonesia (**Fig. 7**). Interestingly, there was a significant lowering of CO
over the Taiwan region in 2016, which agrees with the observed low CO values from the in-situ
measurements at LABS (**Fig. 2b**). Also in agreement, 2016 was a La Niña and negative IOD year
and fire activity was much weaker (**Fig. 2c** and **2d**). During the La Niña years, large-scale
dynamical processes are greatly reversed with respect to El Niño years. We further analyzed the
GpH and wind circulation patterns in 2016 (**Fig. 8**). A significant high-pressure system (western
North Pacific subtropical High) was present over the western North Pacific region in 2016, which



was shifted considerably further eastward compared to over the SCS in 2006 and 2015. The wind
vectors also highlighted the transport of a clean marine air mass from the Pacific Ocean to LABS
in 2016. Interestingly, the vertical pressure velocity exhibited a pronounced upward wind over
Taiwan in 2016, in contrast to the downward wind in 2006 and 2015.  This indicates that dominant
clean marine air reached LABS in 2016 resulting in the lowest CO mixing ratio in the entire dataset
at LABS.



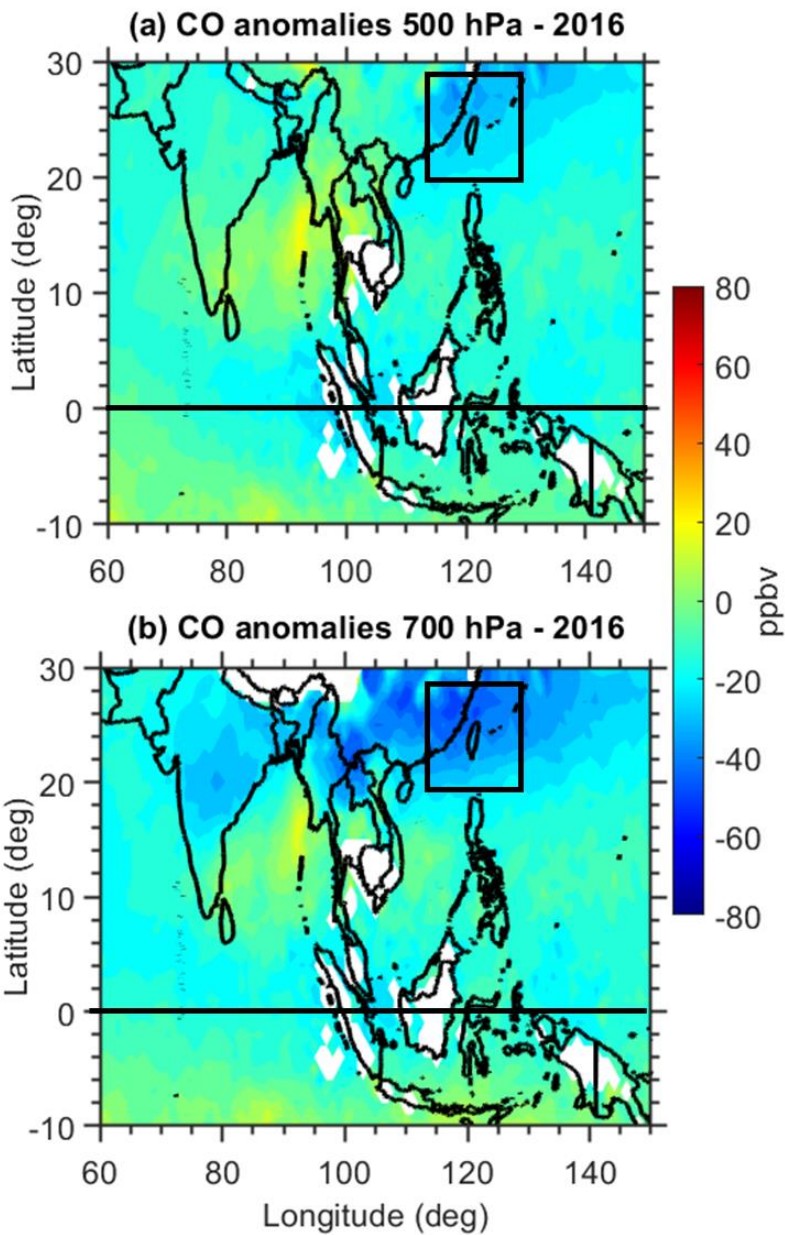


**Figure 7.** Monthly mean CO deviations from the long-term mean (2001-2021) were obtained

from MOPITT satellite observations (a) at 500 hPa and (b) at 700 hPa during October 2016.



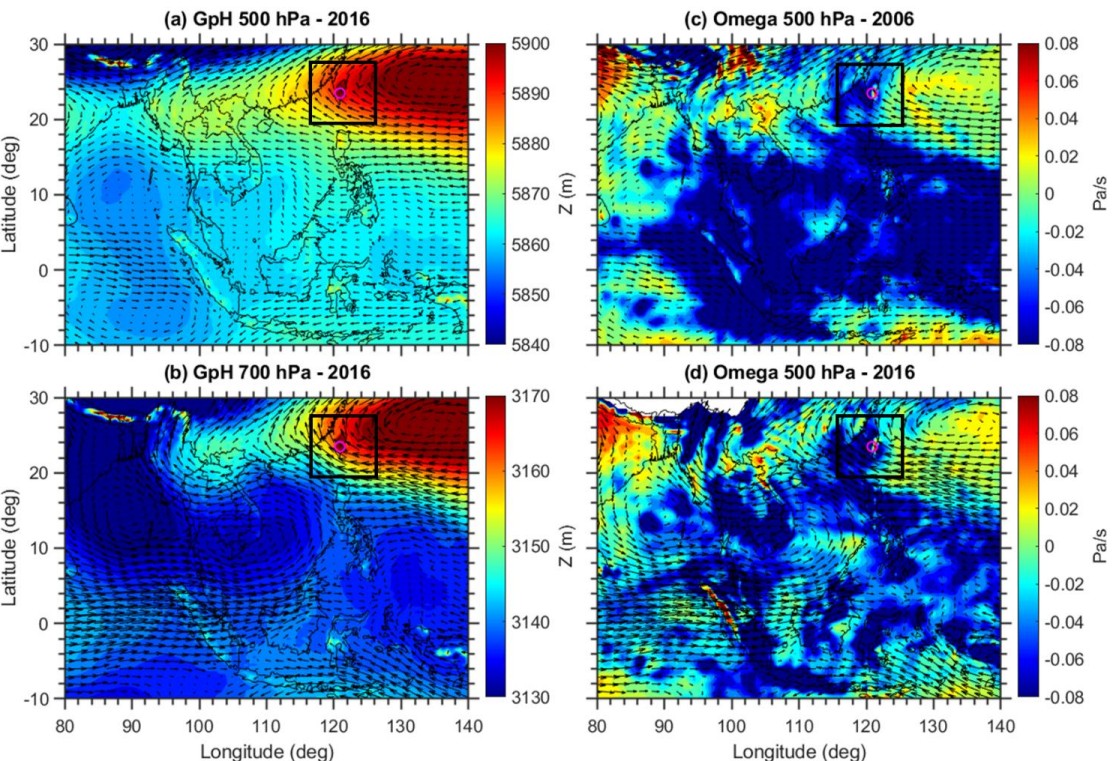


**Figure 8.** Monthly mean Geopotential height (GpH) obtained from MERRA-2 reanalysis (a) at

500 hPa, and (b) at 700 hPa during October 2016. The subplots (c) and (d) are the same as subplots

(a) and (b) but for the observed vertical pressure velocity (Omega).

### 3.4 Role of Hadley circulation

The Hadley circulation (HC) is a crucial component of the climate system, which is

characterized by a thermally driven large-scale meridional circulation (Hadley, 1735). This

circulation links the troposphere and stratosphere and the tropics and extra-tropics, through

horizontal and vertical motions, transporting moisture, heat, and momentum to regulate Earth's

energy budget. As the CO sources (Indonesia) in this study were close to the equator, it is expected

that air tends to rise more or less directly over the CO sources. **Figure 9** shows the vertical-

meridional cross-section of CO and vertical pressure velocity in separate panels averaged along

110°–130°E in October 2006 and 2015. The black-colored vertical line in all the panels in **Figure**

**9** shows the location of LABS and the horizontal line represents the 700 hPa. The vertical cross-

section of CO highlights the uplifting of CO into the upper troposphere over the equator, followed





by southward and northward movement in both 2006 and 2015 (**Fig. 9a** and **9b**). A clear transport
of CO from the source region to the sub-tropics via meridional transport was evident in both events.
It is noted that the higher CO observed between 20–30°N latitude below ∼700 hPa is related to
anthropogenic emissions and not due to the Indonesian fires. To confirm the lofted CO from
Indonesia is really descended in the subtropics due to the Hadley circulation, we looked into the
vertical cross-section of vertical pressure velocity in both events. From Figure 8, it is suggested
that large amounts of CO from Indonesia were transferred into the free troposphere by the strong
upward air motion in this region. Similarly, there was a pronounced descending motion (positive
values of vertical pressure velocity) during October 2006 (**Fig. 9c**) in the northern hemisphere
subtropics around 20–30°N latitude, which corresponds well with the location of LABS. However,
in October 2015, the descending motion was not significant compared to 2006. This may be due
to the different El Niño conditions in 2006 and 2015. While IOD conditions were indeed similar
between 2006 and 2015 (**Fig. 2d**), the higher descending motions in 2006 can be explained in part
by the moderate El Niño conditions during that year. A well-developed El Niño condition was
already established in 2015 compared to 2006. In October 2006, the observed Niño 3.4 value was
around 0.7 whereas in 2015 it was around 2.21. These values indicate that the El Niño conditions
were already well established in October 2015 whereas, in 2006, the conditions were not developed
as El Niño. It is reported that in El Niño conditions, the western Pacific HC is observed to be
weakened whereas the eastern Pacific HC is strengthened (Wang, 2004). This is supported by the
observed lesser descending motions in 2015 from the present study. These differences in the
descending motions likely influenced the greater CO enhancement in 2006 compared to 2015 at
LABS (**Fig. 2b** and **Table 1**). Overall, it is clearly illustrated from the MOPITT CO vertical cross-
section and the MERRA-2 vertical pressure velocity that the CO emitted from the Indonesian fire
was transported vertically through the Hadley circulation to the LABS location.

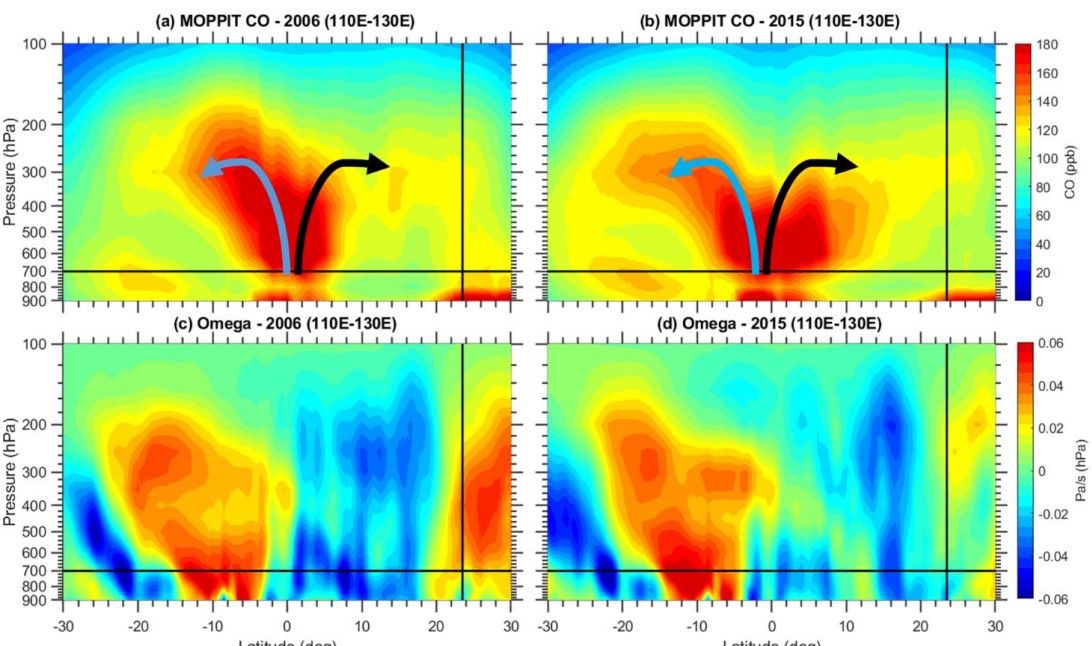

**Figure 9**. Vertical–meridional cross-section of MOPITT CO averaged along 110°–130°E (a) for October 2006 and (b) for October 2015. Subplots (c) and (d) are the same as subplots (a) and (b) but for the MERRA-2 reanalysis vertical pressure velocity. Positive (negative) values represent the downward (upward) wind.

## 4. Summary and Conclusions

Changes in the background climate will inevitably impact meteorological transport processes and the concentrations of pollutants arriving at downwind regions. Lulin Atmospheric Background Station (LABS, 23.47°N 120.87°E, 2862 m ASL) is the only high-altitude background station located in the western North Pacific region, and is optimally located to study some of these transport processes, including long-range transport of pollution in the free troposphere and stratospheric intrusions. During October 2006 and 2015, there were substantial increases in CO mixing ratios, ~47.2 ppb (37.2%) and ~36.7 ppb (28.9%) increase compared to the 16-year (2006-2021) means at LABS. Interestingly, these two events (2006 and 2015) were strongly associated with the two major biomass burning episodes over Indonesia, which resulted from a combined impact of positive phase ENSO and IOD-induced drought conditions. MODIS active fire counts showed the largest fires in October 2006 and 2015 compared to the other years in the 16-year



period in Indonesia. These record fires reflected two of the largest carbon emissions in the
Indonesian region since 1997. Apart from these high values in October 2006 and 2015, in October
2016, extremely low CO values were recorded at LABS (~31.4% lower compared to 2006-2021
mean). October 2016 was associated with negative IOD over the Indian Ocean and La Niña in the
Pacific Ocean, resulting in the lowest fire activity over the MC. Further, we found that the large-
scale circulations in 2016 were quite different from 2006 and 2015. In 2016, LABS was dominated
by the southerlies due to the western Pacific subtropical High (WPSH), which transported clean
marine air from the Pacific Ocean and caused record low CO values at LABS. The main aim of
our study was to examine the transport pathways of CO from Indonesia's source region to the
downwind LABS region. By comparing the CO and atmospheric circulation data from the 2006
and 2015 El Niño (positive IOD) years and 2016 La Niña (negative IOD), we found two plausible
transport pathways of CO from Indonesia to LABS.
**Figure 10** illustrates a schematic diagram of the major transport pathways of CO from
Indonesia to subtropical East Asia during the two event years. They include horizontal transport
in the free troposphere due to El Niño-induced high-pressure anticyclone circulation and vertically
through the Hadley circulation. For October 2006 and 2015, corresponding to El Niño and positive
IOD, northern SCS was influenced by the high-pressure anti-cyclonic system in the free
troposphere.  The southerlies on the southwest flank of the anticyclone further merged with the
subtropical westerlies over PSEA and then transported polluted air to LABS. Apart from this
horizontal transport, CO was transported through the Hadley circulation to LABS in both events.
However, there was a distinctly different HC strength in 2006 compared to 2015 due to the
different El Niño conditions. These two events were strongly associated with positive IOD, but in
2006, the El Niño conditions were not well developed, whereas in 2015 well-developed El Niño
conditions were evident. These El Niño conditions further suppressed the HC over the western
Pacific in 2015 compared to 2006. This suggested the importance of the background climate
conditions (ENSO and IOD) on the pollutant transport process. A changing warmer climate can
influence carbon emissions and alter the transport pathways, hence impacting the various scales of
air pollution and climate. Overall, the present results further provide knowledge to the atmospheric
chemistry community about the different transport pathways of pollutants and the role of climate
conditions.



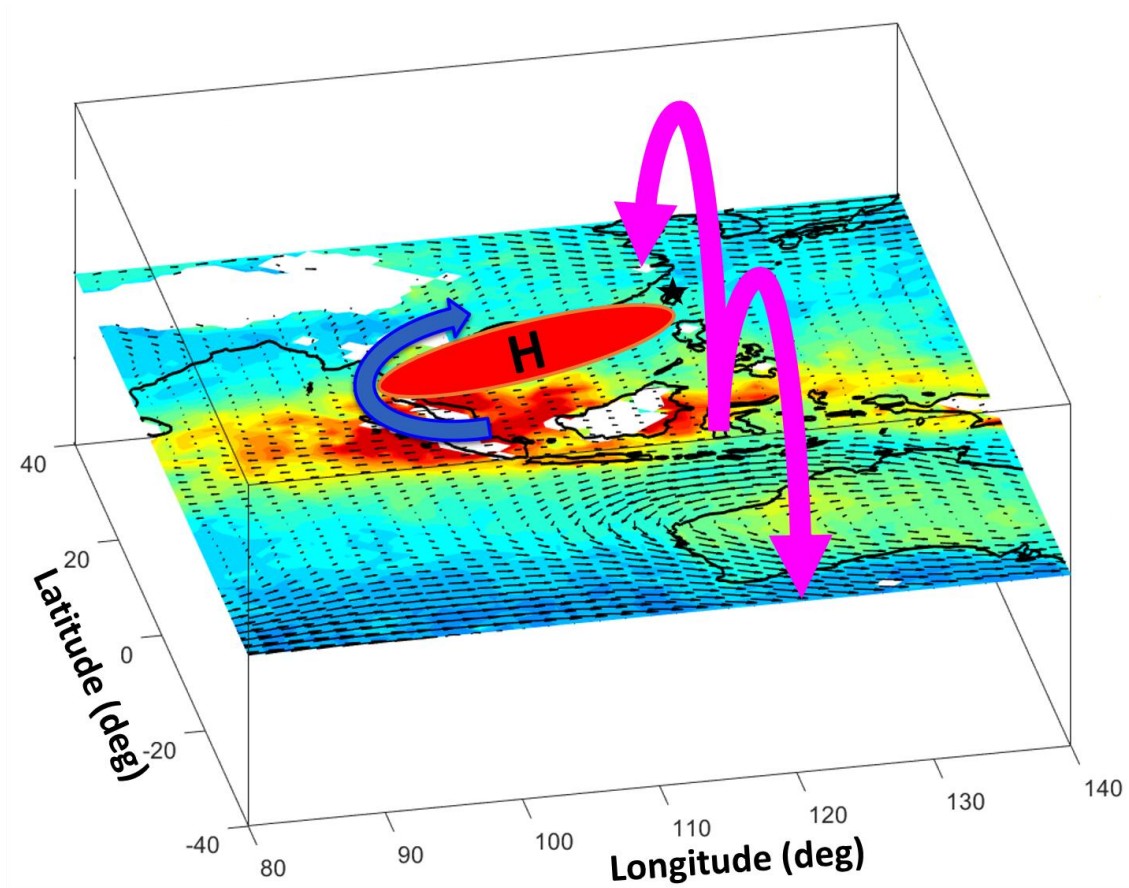

**Figure 10.** Schematic diagram of CO transport from Indonesian fires to subtropical East Asian region. Horizontal transport of CO due to the high-pressure anticyclone is denoted by the blue-colored arrow. H denotes high-pressure anticyclone over northern parts of the South China Sea. Magenta-colored arrows indicate the transport of CO through the local Hadley circulation (over 110°–130°E). Black-colored star symbol represents the LABS location.

**Data availability**

The CO data at LABS can be assessed at http://lulin.tw/index_en.htm. The AIRS and MOPITT CO data can be downloaded from the following websites https://disc.gsfc.nasa.gov/datasets/AIRS3STM_7.0 (AIRS project., 2019) and https://asdc.larc.nasa.gov/project/MOPITT. MERRA-2 data are available online through the



NASA Goddard Earth Sciences Data Information Services Center (GES DISC;
https://disc.gsfc.nasa.gov, last access: 30 May 2022). Nino 3.4 Index and IOD data can be
downloaded through the following websites https://psl.noaa.gov/gcos_wgsp/Timeseries/Niño34/.
https://psl.noaa.gov/gcos_wgsp/Timeseries/DMI/. The MODIS fire products can be downloaded
from the following website https://firms.modaps.eosdis.nasa.gov/active_fire/.
**Author contributions**
**Saginela Ravindra Babu:** Conceptualization, Data curation, Formal analysis, Investigation,
Software, Validation, Visualization, Writing – original draft preparation, Writing – review and
editing; **Chang-Feng Ou-Yang**: Data curation, Software, Validation, Visualization; **Stephen M.**
**Griffith**; Writing – review and editing; **Shantanu Kumar Pani**: Data curation and Visualization;
**Steven S. Kong**: Data curation and Visualization; **Neng-Huei Lin**: Conceptualization,
Investigation, Funding Acquisition, Supervision, Resources, Writing – review and editing.
**Competing Interest**
The authors declare that they have no conflict of interest.
**Acknowledgments**
The work is primarily supported by the Ministry of Science and Technology, Taiwan under the
grants of MOST 110-2811-M-008-562 and MOST 109-2811-M-008-553. Authors thanks to
Taiwan Environmental Protection Administration (TEPA) for supporting the air pollutants
monitoring at LABS. The authors thank NASA and NOAA for providing MOPITT, MODIS, and
AIRS satellite data. We thank NASA's Global Monitoring and Assimilation Office (GMAO) for
providing the Modern-Era Retrospective analysis for Research and Applications, Version 2
(MERRA-2) data. We also thank NOAA ESRL Physical Sciences Laboratory for providing Indian
Ocean Dipole and Niño 3.4 index values through the following websites
https://psl.noaa.gov/gcos_wgsp/Timeseries/DMI/
https://psl.noaa.gov/gcos_wgsp/Timeseries/Niño34/.







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



**Table 1**. Detailed statistics of observed CO in October during 2006 to 2021 at LABS.

| Year | Mean | Median | Standard Deviation | Change in CO (%) | Total data points |
|------|------|--------|--------------------|------------------|-------------------|
| **2006** | **175.8** | **174** | **51** | **33.9** | **703** |
| 2007 | 155.3 | 140 | 63.4 | 18.3 | 732 |
| 2008 | 125.5 | 125 | 26.9 | -4.4 | 599 |
| 2009 | 127.1 | 125 | 35.5 | -3.2 | 533 |
| 2010 | 143.9 | 136 | 38.1 | 9.6 | 739 |
| 2011 | 137.1 | 137 | 41.9 | 4.4 | 734 |
| 2012 | 155.8 | 153 | 39.4 | 18.7 | 643 |
| 2013 | 146.8 | 141 | 35.7 | 11.8 | 365 |
| 2014 | 125.6 | 120 | 39.8 | -4.2 | 602 |
| **2015** | **164.8** | **163.5** | **46.2** | **25.6** | **732** |
| 2016 | 91.6 | 87 | 20.9 | -30.2 | 732 |
| 2017 | 109.7 | 100.3 | 32.4 | -16.4 | 744 |
| 2018 | 147.7 | 149.9 | 29.1 | 12.5 | 736 |
| 2019 | 142.4 | 142.8 | 37.7 | 8.5 | 742 |
| 2020 | 121.3 | 113.8 | 29.5 | -7.5 | 742 |
| 2021 | 107.7 | 104.6 | 26.9 | -17.9 | 744 |
