# Peer review of "Transport pathways of carbon monoxide from Indonesian fire pollution to a subtropical high-altitude mountain site in western North Pacific"

_Atmospheric Chemistry and Physics, 2022_

## Referee Comment (RC2)

**Review report of the manuscript "Transport pathways of carbon monoxide from Indonesian fire pollution to a subtropical high-altitude mountain site in western North Pacific".**

**General comments:**

This is an interesting study looking at the long range transport impacts on the observed enhancements in pollutants at the high altitude pollution monitoring station in Taiwan. It explains the challenges and significance of air pollution events which will disperse to a larger area. However the manuscript needs some improvement to bring more clarity on the study. Specific points of concern are given below. In general English also needs a significant improvement (some in notified, but many I did not). This work can be accepted for the publication after the successful revision of the following points.

**Specific concerns:**

**Lines 44-45:** How do both extreme and weak El Niño events relate with forest fires, what is the basis here? This looks little contrasting to me. Weak El Niño should not result in extreme dry conditions and that would be non-conducive for fire events.

**Lines 50-51:** Please check the grammar the sentence is grammatically not correct.

**Line 51:** Check the grammar

**Lines 64 and 65:** The average atmospheric life time of CO is two months and that of $CH_4$ is close to 12 years. Any episodic increase in CO may not have direct impact on average $CH_4$ atmospheric life time through OH radical chemistry because CO is prone for more local variations and so are OH radicals. Only sustained increase of CO in all the regions may cause that effect but it is very vague to state that, CO increase may increase the $CH_4$ life time through OH processing. Kindly bring more clarity on this statement.

**Line no. 74:** I would suggest to give the site details separately. It has been merged with introduction which does not sync. You can revise the introduction by keeping the studies reported from LABS and objectives for this study. Bring out the site details along with more details on local meteorology in a separate section. Local meteorology at the study site is missing and would be needed for the reader to understand your results.

**Line 79:** "The LABS is often found within the free troposphere". Again the statement is very vague. Please try to show the ABL height in reference to the station height for different seasons. The ABL height plays an important role in the interpretation of long range transport and local emissions. Measurements of CO experience boundary layer local emission effects if the station is within boundary layer. This should be considered carefully while deciding the effect of long-range transport. Further the site looks to be in between dense forest region how do you remove the local forest fire event effects from the long range transport from Indonesia?

**Lines 102-103:** What is the CO trend in 16 years? Have you considered the trend while estimating the enhancement during 2006 and 2015? Because long term CO may have natural variability (de-seasoned) in its mean and that needs to be removed while calculating the enhancement.

**Line 129:** first line indent is not followed here.

**Figure 2**: What is the natural trend of CO over the years removing the episodic events? How do you separate the natural viability with episodic enhancements due to forest fires? If the more fire activity is bringing more CO then why year 2014 has not shown any enhancement even through fire activity and Niño 3.4 are comparatively high. Same is the case in the year 2009.

**Figure 3**: This correlation is drawn for which pressure level of satellite data?. In situ measurements are point measurements at the surface whereas, satellite data are area averaged and column integrated. If the columnar area averaged data are used will it represent the true scenario of LABS? And the further interpretation of enhancement is logical? The clarity is missing here. This is important because satellite may have picked the local fire event enhancements too. It would be better to incorporate a fire event intensity distribution diagrams (for the years 2006 and 2015) around the LABS site and then overlap air mass trajectories (use of polar plots may help) receiving at the site to see the real influence of the detected fire events. This should normally correlate with the enhancement. After establishing this relationship dynamics can be explained.

**Lines 218-219:** Did you subtract the 2006 and 2015 data from long term mean of MOPITT CO observations or the other way around? You were looking for the enhancements, then long term mean should be subtracted from 2006 and 20015 data to see the magnitude of enhancement. Please re-verify this statement.

**Line 224:** What is the uncertainty of MERRA-2 reanalysis data and how significant it is in your interpretation of impact of GpH and wind distribution while explaining the transport pathways?

Lines 290-291: What is the time scale of CO transportation form the source region to the observational site via meridional transport?. Whether it fits observed changes?

**Summary and Conclusions:** It looks more like a discussion rather than conclusion. Please bring crisp 4-5 salient points of this study in conclusion. Figure 10 and related content can go as a discussion. It does not sync again in summary and conclusion.

---

## Author Comment (AC1)

**Response to Reviewers:**

The authors greatly appreciate the reviewers' constructive comments to further improve our manuscript's quality. We carefully considered each comment and revised our manuscript to address the issues raised. The original reviewer comments are in black and our replies are in blue. Text excerpts are italicized in *blue* with new text in *bold*.

**Response to Reviewer #1**

This a very interesting and well-structured paper. It shows the transport pathways of CO from Indonesia to sub-tropical high-altitude locations during two extreme fire pollution events (2006 and 2015) using in-situ and satellite measurements along with MERRA-2 reanalysis products. The topic of this study is interesting and the authors have presented the results with sufficient analyses. The manuscript could be considered to be published in ACP after the following revision.

**Reply**: We wish to thank the reviewer for their review of our paper and for appreciating the content of the manuscript. We have revised the manuscript while considering the reviewer's comments/suggestions.

I have two major suggestions/comments for the authors

Before Figure 2, in the manuscript, the authors could provide a vertical cross-section of CO over the maritime continent from satellite measurements. This will give a better understanding of CO inter-annual variability and the high CO enhancement in two events, particularly in October 2006 and 2015 compare to other years.

**Reply**: Thanks for the constructive comment. In the revised manuscript, we have included the above-mentioned plots as Figure 2. These plots show a height-time cross-section of CO observed over the Maritime Continent from January 2003 to 2021 obtained from AIRS (top) and MOPITT (bottom) satellite measurements.

**Figure2**. Height-time cross-section of CO observed over the Maritime continent (average over 90E-140E,10S-10N) during 2003-2021 obtained from (a) AIRS, and (b) MOPITT satellite measurements.

As mentioned in the introduction by the authors, the 1997 fire event was one of Indonesia's worst fire events. Are there any similarities between 1997 and 2006 and 2015, particularly in large-scale circulations? It would be great if the authors add the large-scale circulations in October 1997, before the conclusions section.

**Reply**: Thanks for the nice suggestion. We agree with the reviewer that 1997 was an extreme El Niño event and had a strong impact on the global climate. As suggested by the reviewer, we checked the large-scale circulations in the 1997 event by using MERRA-2 reanalysis products and found a quite similar circulation characteristic as noticed in the 2006 and 2015 events. In the revised manuscript, we have included the above-mentioned plots as Sup. Figure4. Please find the below figure.

---

## Author Comment (AC2)

Response to Reviewers:

The authors greatly appreciate the reviewers' constructive comments to further improve our manuscript's quality. We carefully considered each comment and revised our manuscript to address the issues raised. The original reviewer comments are in black and our replies are in blue. Text excerpts are italicized in *blue* with new text in ***bold***.

Response to **Reviewer #2**

This is an interesting study looking at the long range transport impacts on the observed enhancements in pollutants at the high altitude pollution monitoring station in Taiwan. It explains the challenges and significance of air pollution events which will disperse to a larger area. However, the manuscript needs some improvement to bring more clarity on the study. Specific points of concern are given below. In general English also needs a significant improvement (some in notified, but many I did not). This work can be accepted for the publication after the successful revision of the following points.

**Reply:** We highly appreciate the thoughtful and valuable suggestions by the reviewer, which are helpful for us to improve the quality of our manuscript. We have revised the manuscript with consideration of the reviewer's comments/suggestions. We have taken utmost care in the revised manuscript about English grammar and usage. The revised manuscript was thoroughly checked by English native speaker (SG; one of the co-author in the manuscript).

Specific concerns:

Lines 44-45: How do both extreme and weak El Niño events relate with forest fires, what is the basis here? This looks little contrasting to me. Weak El Niño should not result in extreme dry conditions and that would be non-conducive for fire events.

**Reply:** Based on the value of Nino 3.4, the 2006 El Niño was weak compared to the 2015 event. However, the Indian Ocean Dipole (IOD) was in a positive phase in both events and played an important role in causing dry conditions over the Maritime Continent. The roles of IOD and El Niño in fire activity over the Maritime continent have been well reported (Please see Pan et al., 2018 for more details). In, 2006, the combination of positive IOD and weak or moderate El Nino conditions impacted the fire activity. To avoid confusion, we have modified the sentence in the revised manuscript.

Please refer to Lines 44-46:

***"For example, dry conditions associated with the positive IOD during the 2015/16 El Niño and 2006/07 El Niño events led to increased fire activity over Indonesia and the wider MC."***

Lines 50-51: Please check the grammar the sentence is grammatically not correct.

Line 51: Check the grammar

**Reply**: Corrected in the revised manuscript. Please refer to Lines 50-52:

*"The impact of these two Indonesian fire events on carbon emissions, tropospheric trace gases, aerosol composition, and air quality has been extensively discussed in the literature."*

Lines 64 and 65: The average atmospheric life time of CO is two months and that of CH4 is close to 12 years. Any episodic increase in CO may not have direct impact on average CH4 atmospheric life time through OH radical chemistry because CO is prone for more local variations and so are OH radicals. Only sustained increase of CO in all the regions may cause that effect but it is very vague to state that, CO increase may increase the CH4 life time through OH processing. Kindly bring more clarity on this statement.

**Reply:** We have modified the sentence in the revised manuscript. Please refer to Lines 67-69:

*"CO is also an ozone ($O_3$) precursor in the troposphere, and indirectly increases radiative forcing (0.23 +/- 0.05 W m−2) through the production of $O_3$ and $CO_2$ and depletion of hydroxyl radical, the primary chemical reactant with $CH_4$ in the atmosphere (IPCC, 2013)."*

Line no. 74: I would suggest to give the site details separately. It has been merged with introduction which does not sync. You can revise the introduction by keeping the studies reported from LABS and objectives for this study. Bring out the site details along with more details on local meteorology in a separate section. Local meteorology at the study site is missing and would be needed for the reader to understand your results.

**Reply:** Thanks for the nice suggestion. We have included the site details separately (Sec. 2.1) in the revised manuscript. Details of the various meteorological measurements at LABS have been previously described in detail (Sheu et al., 2009; Ou-Yang et al., 2014; Ravindra Babu et al., 2022) and are thus only briefly described in the present study. We included the meteorological conditions at the study location based on in-situ measurements from 2006 to 2021 and provided them in the supplementary figures. The figure (sup. Figure 1 in the revised manuscript) below shows the climatological monthly mean of various meteorological parameters at LABS along with the MERRA-2 boundary layer height around LABS.

[Figure]

**Figure S1**. Long-term monthly mean of (a) temperature, (b) relative humidity, (c) wind speed, (d) wind direction, (e) carbon monoxide at LABS, and (f) MERRA-2 obtained boundary layer height around LABS between 2006 and 2021. Vertical error bars indicate the standard deviation from the monthly mean.

Line 79: "The LABS is often found within the free troposphere". Again the statement is very vague. Please try to show the ABL height in reference to the station height for different seasons. The ABL height plays an important role in the interpretation of long range transport and local emissions. Measurements of CO experience boundary layer local emission effects if the station is within boundary layer. This should be considered carefully while deciding the effect of long-range transport. Further the site looks to be in between dense forest region how do you remove the local forest fire event effects from the long range transport from Indonesia?

**Reply:** ABL height information is not available at LABS from in-situ measurements. However, the MERRA-2 boundary layer height around LABS was obtained between 2001 to 2021 and plotted along with the various meteorological parameters at LABS. Please see Figure S1f above.

Regarding local fire activity around the study location, we further checked the MODIS fire counts over Taiwan during the 2006 and 2015 events. Please see the attached Figure R4 for the spatial distribution of MODIS fire counts over Taiwan. It is very clear that the local fire activity around the study location was negligible in both events. We also compared the total fire counts in Indonesia with the total fire counts in Taiwan in both events. For example, the total number of

MODIS fire counts for Indonesia on October 2006 is >40000, whereas it is only 9 for Taiwan. Similarly, in October 2015, the total MODIS fire counts in Indonesia was >50000 whereas for Taiwan it was only 3. Also, the fire counts were mostly having confidence level below 80 in both events (see the Table R1).

[Figure]

**Figure R1.** MODIS fire hot spots are shown as red dots on (a) October 2006, and (b) October 2015. Magenta-colored star symbol represents the LABS location.

Table R1. Details of MODIS fire counts during October 2006 and 2015.

| Latitude | Longitude | Day number | Year | Confidence |
|---|---|---|---|---|
| 23.9631 | 120.967 | 3 | 2006 | 64 |
| 24.3039 | 121.416 | 7 | 2006 | 58 |
| 23.7057 | 120.322 | 14 | 2006 | 77 |
| 22.5443 | 120.3525 | 15 | 2006 | 62 |
| 23.4975 | 120.1791 | 18 | 2006 | 59 |
| 22.5383 | 120.3544 | 22 | 2006 | 64 |
| 23.4946 | 121.3367 | 23 | 2006 | 39 |
| 24.856 | 120.9863 | 25 | 2006 | 53 |
| 23.9743 | 120.6893 | 29 | 2006 | 58 |
|  |  |  |  |  |
| 23.7126 | 121.4815 | 3 | 2015 | 57 |
| 23.8096 | 121.5147 | 5 | 2015 | 43 |
| 22.5404 | 120.3508 | 8 | 2015 | 78 |
| 25.0337 | 121.177 | 13 | 2015 | 53 |

Lines 102-103: What is the CO trend in 16 years? Have you considered the trend while estimating the enhancement during 2006 and 2015? Because long term CO may have natural variability (deseasoned) in its mean and that needs to be removed while calculating the enhancement.

**Reply:** Yes, we agree with the reviewer that there might be natural variability in CO data. However, we have subtracted 16-year mean data from 2006 and 2015 individual data. So any natural variability will be nullified. There is decreasing trend in CO at LABS during last 16 years.

Line 129: first line indent is not followed here.

**Reply:** Corrected in the revised manuscript. Please refer to Lines 160-162:

**"*We also utilized monthly mean geopotential height (GPH), wind vectors (zonal and meridional wind speed), and pressure vertical velocity from the Modern-Era Retrospective Analysis for Research and Applications, version 2 (MERRA-2).*"**

Figure 2: What is the natural trend of CO over the years removing the episodic events? How do you separate the natural viability with episodic enhancements due to forest fires? If the more fire activity is bringing more CO then why year 2014 has not shown any enhancement even through fire activity and Niño 3.4 are comparatively high. Same is the case in the year 2009.

**Reply:** We have subtracted 16-year mean data from 2006 and 2015 individual data. So any natural variability will be nullified. At LABS, we observed decreasing trend in CO during 2006 to 2021.

Please see Figure 2 in the revised manuscript. The height-time cross-section of CO over the Maritime Continent (MC) clearly shows the extreme CO values in 2006 and 2015. Even though 2009 and 2014 were El Niño years, the CO over MC was not high as observed in 2006 and 2015. The weaker and shorter duration of fire activities could largely explain the less CO over MC in 2009 and 2014 in contrast to those in 2006 and 2015.

Figure 3: This correlation is drawn for which pressure level of satellite data?. In situ measurements are point measurements at the surface whereas, satellite data are area averaged and column integrated. If the columnar area averaged data are used will it represent the true scenario of LABS? And the further interpretation of enhancement is logical? The clarity is missing here. This is important because satellite may have picked the local fire event enhancements too. It would be better to incorporate a fire event intensity distribution diagrams (for the years 2006 and 2015) around the LABS site and then overlap air mass trajectories (use of polar plots may help) receiving at the site to see the real influence of the detected fire events. This should normally correlate with the enhancement. After establishing this relationship dynamics can be explained.

**Reply:** We are sorry for not mentioning the pressure level which we used CO data from the satellite measurements. Actually, we used 700 hPa (close to LABS's altitude) CO data from both satellites and made correlations in the present study. We have included this in the revised manuscript.

We also checked the fire hot spots over Taiwan in October 2006 and 2015 from MODIS fire products. Please see the attached Figure R4 for clarity. It is very clear that there is negligible fire activity over entire Taiwan in both events. Also, there is no fire activity near the LABS location in both events. This clearly indicates there is no local fire activity impact on CO measurements at

LABS in both events. Also, the background circulations from the present study (see the manuscript) clearly supported the long-range transport of CO from the MC to the LABS location.

Lines 218-219: Did you subtract the 2006 and 2015 data from long term mean of MOPITT CO observations or the other way around? You were looking for the enhancements, then long term mean should be subtracted from 2006 and 20015 data to see the magnitude of enhancement. Please reverify this statement.

**Reply:** Yes, long-term mean was subtracted from the 2006 and 2015 data.

Line 224: What is the uncertainty of MERRA-2 reanalysis data and how significant it is in your interpretation of impact of GpH and wind distribution while explaining the transport pathways?

**Reply**: Reanalysis products are the result of the assimilation of observations from different sources into an atmospheric model that generates evenly distributed global data. MERRA-2 (Molod et al 2015) is the most recent reanalysis produced by NASA's Global Modelling and Assimilation Office (GMAO). It uses the Goddard Earth Observing System-5 (GEOS-5) atmospheric general circulation model (AGCM) with a 4D-VAR data assimilation scheme. We are not aware of the uncertainty of MERRA-2 reanalysis data.

   To clarify this, we cross-checked the geopotential height and wind data from NCEP-DOE Reanalysis-2 and ERA-5. Figure R2 shows the monthly mean Geopotential height (GpH) obtained from (a) NCEP/DOE Reanalysis II, (b) ERA-5, and (c) MERRA-2 reanalysis for October 2006. The three reanalysis shows quite a similar pattern in GpH (presence of anti-cyclonic circulation over the South China Sea) and wind pattern in October 2006. Even if there is uncertainty as raised by the reviewer in GpH data from MERRA-2, it will not affect our main results in the present study. All the reanalysis GpH and wind patterns clearly indicated the presence of anti-cyclonic circulation with a high-pressure system over the south china sea in October 2006. This provides us with strong confidence in our results. Please see the below attached Figure R2, respectively.

[Figure]

**Figure R2**. Monthly mean Geopotential height (GpH) obtained from (a) NCEP/DOE Reanalysis II, (b) ERA-5, and (c) MERRA-2 reanalysis for October 2006.

Lines 290-291: What is the time scale of CO transportation form the source region to the observational site via meridional transport?. Whether it fits observed changes?

**Reply:** Based on available observations from the present study, it is quite difficult to tell the time scale for transportation from the source region to the receptor site. It needs more detailed modeling

and numerical simulations. In this work, our major goals are to investigate the plausible transport pathways of CO from the maritime continent to sub-tropical high-altitude locations. In future studies, we will look at this interesting question raised by the reviewer.

Summary and Conclusions: It looks more like a discussion rather than conclusion. Please bring crisp 4-5 salient points of this study in conclusion. Figure 10 and related content can go as a discussion. It does not sync again in summary and conclusion.

**Reply:** We have modified the conclusion section as suggested.

References

Pan, X., Chin, M., Ichoku, C. M., and Field, R. D.: Connecting Indonesian fires and drought with the type of El Niño and phase of the Indian Ocean dipole during 1979–2016, J. Geophys. Res.-Atmos., 123, 1–15, https://doi.org/10.1029/2018JD028402, 2018.

Sheu, G.-R., Lin, N.-H., Wang, J-L., Lee,C-T.; Lulin Atmospheric Background Station: A New High-Elevation Baseline Station in Taiwan, J-STAGE, Volume 24, Issue 2, Pages 84-89, https://doi.org/10.11203/jar.24.84, 2009.

Ou-Yang, C. F., Lin, N. H., Lin, C. C., Wang, S. H., Sheu, G. R., Lee, C. Te, Schnell, R. C., Lang, P. M., Kawasato, T. and Wang, J. L.: Characteristics of atmospheric carbon monoxide at a high-mountain background station in East Asia, Atmos. Environ., 89, 613–622, https://doi.org/10.1016/j.atmosenv.2014.02.060, 2014.

Ravindra Babu, S., Pani, S.K., Ou-Yang, C.F., Lin, N.H.: Impact of 21 June 2020 Annular Solar Eclipse on Meteorological Parameters, O3 and CO at a High Mountain Site in Taiwan. Aerosol Air Qual. Res. 22, 220248. https://doi.org/10.4209/aaqr.220248, 2022.

---

## Author Response (AR2)

Response to technical corrections for "**Transport pathways of carbon monoxide from Indonesian fire pollution to a subtropical high-altitude mountain site in western North Pacific**" by Saginela RavindraBabu et al.

Dear Editor,
Jayanarayanan Kuttippurath,

We are submitting our technically corrected revised article titled "**Transport pathways of carbon monoxide from Indonesian fire pollution to a subtropical high-altitude mountain site in western North Pacific**". The authors wish to thank the Editor for taking care of our manuscript.

Please noted that we have made one relevant change in the final revised manuscript. We have included **Sup. Figure 4** as **Figure 10** in the final revised manuscript.

**Response to technical corrections**

1. Coloured or marked text in *.pdf manuscript file is not allowed. Please provide a clean version of *pdf manuscript file (with black text) with the next revision.

**Reply:** We provided a clean version of the manuscript with black text in the revised version.

2. For the next revision, please check if your figures containing photos require a copyright statement/image credit and add it to the figures (or captions) (https://publications.copernicus.org/for_authors/manuscript_preparation.html#figurest ables -> Reproduction and reuse of figures and tables). If these figures were entirely created by the authors, there is no need to add a copyright statement or credit. In that case it is important that you confirm this explicitly by email.

**Reply:** Yes, the figures in the manuscript (photos) were entirely created by the authors themselves. It is noted that one of the corresponding authors (Neng-Huei Lin) is a PI (Principal investigator) for the Lulin Atmospheric Background Station (LABS) project and all the co-authors are team members of the LABS project. As a part of the project, we have taken these photos by ourselves.

3. With the next revision, please re-number the figures of the manuscript consecutively. It seems that Figure 7 is missing from the manuscript.

**Reply:** We are sorry for the mistake. We have corrected and renumbered the figures in the revised manuscript.

**We once again thank the Reviewers and Editor for going through the manuscript carefully and providing constructive comments/suggestions which made us improve the manuscript content significantly.**